# Cyber-Attack Scoring Model Based on the Offensive Cybersecurity Framework

**Kyounggon Kim** [1,*] **, Faisal Abdulaziz Alfouzan** [2,*] **and Huykang Kim** [3]

1 Center of Excellence in Cybercrime and Digital Forensics, College of Criminal Justice, Naif Arab University for Security Sciences, Riyadh 14812, Saudi Arabia

2 Department of Forensic Sciences, College of Criminal Justice, Naif Arab University for Security Sciences, Riyadh 14812, Saudi Arabia

3 School of Cybersecurity, Korea University, Seoul 02841, Korea; cenda@korea.ac.kr

* Correspondence: kkim@nauss.edu.sa (K.K.); falfouzan@nauss.edu.sa (F.A.A.)

**Abstract:** Cyber-attacks have become commonplace in the world of the Internet. The nature of cyber-attacks is gradually changing. Early cyber-attacks were usually conducted by curious personal hackers who used simple techniques to hack homepages and steal personal information. Lately, cyber attackers have started using sophisticated cyber-attack techniques that enable them to retrieve national confidential information beyond the theft of personal information or defacing websites. These sophisticated and advanced cyber-attacks can disrupt the critical infrastructures of a nation. Much research regarding cyber-attacks has been conducted; however, there has been a lack of research related to measuring cyber-attacks from the perspective of offensive cybersecurity. This motivated us to propose a methodology for quantifying cyber-attacks such that they are measurable rather than abstract. For this purpose, we identified each element of offensive cybersecurity used in cyber-attacks. We also investigated the extent to which the detailed techniques identified in the offensive cyber-security framework were used, by analyzing cyber-attacks. Based on these investigations, the complexity and intensity of cyber-attacks can be measured and quantified. We evaluated advanced persistent threats (APT) and fileless cyber-attacks that occurred between 2010 and 2020 based on the methodology we developed. Based on our research methodology, we expect that researchers will be able to measure future cyber-attacks.

**Keywords:** offensive cybersecurity; cyber-attacks; scoring model; offensive cybersecurity framework

## 1. Introduction

The development of internet technology has increased the impact of cyber-attacks. The targets of cyber-attacks are steadily changing from traditional systems to cyber–physical systems (CPS). Cyber-attacks on smart mobility and smart homes are steadily increasing at a quickening pace. In 2005, two security researchers revealed the critical vulnerabilities of a self-driving car [1]. They remotely controlled the key features of a self-driving Jeep vehicle and succeeded in stopping the car on a highway. Cyber-attack techniques are becoming more sophisticated and destructive. Not only individual hackers but also state-sponsored hackers are actively entering the field of cyber-attacks. Cyber attackers use offensive cybersecurity technology to perform complex attacks. Offensive cybersecurity refers to a hacking technique that attacks a system, not a defense technology [2]. State-sponsored hackers and the Advanced Persistent Threat (APT) group also use offensive cybersecurity technology.

Attacks on Internet of Things (IoT) and smart homes are also constantly occurring. The Mirai botnet, known as the first IoT malware, attacked the Dyn network server, which is mainly operated in the United States [3]. This prevented Twitter, PayPal, and a significant portion of major online services from providing their services because of the huge amount of network traffic. The attack was not launched from a personal computer such as a zombie

botnet. The Dyn server received a massive load of traffic from IoT devices, and a variety of IoT cam companies were compromised by this malware. Korean offensive security researchers found critical vulnerabilities in Z-Wave, a wireless communication protocol that is commonly used in smart homes to communicate between the gateway and small nodes such as door-lock, multi-tab, and gas-lock [4]. A variety of threats to which smart homes are exposed were investigated by using existing offensive cyber security research methods. Much research regarding cyber-attacks has been conducted; however, there is a lack of research regarding systematic measurement of cyber-attacks.

It is necessary to identify the cyber threat actors to measure a cyber-attack. Cyber threat actors include individual hackers, cyber-terrorists, hacktivists, cybercriminals, and state-sponsored hackers. State-sponsored hackers and cybercrime organizations utilize APT that contains various offensive cybersecurity techniques. Table 1 presents some examples of nations and APT groups that have conducted cyber-attacks [5]. Among these APT analysis surveys, many reports use the terms of "sophisticated" attacks. In this paper, we propose an offensive cybersecurity framework as a method to systematically measure a score for the cyber-attacks in each isolated event. To the best of our knowledge, there have been no studies that score cyber-attacks. Hence, we analyze the degree of cyber-attack techniques for APT and fileless cyber-attacks that are using techniques contained in the offensive cybersecurity framework.

**Table 1.** Names of various Advanced Persistent Threat (APT) groups (sample).

| Nations | APT Groups |
| --- | --- |
| China | APT1, Common Crew, PLA Unit 61398, Group 3, APT2, PLA Unit 61486, Buckeye, Gothic Panda |
| Russia | Sofacy, APT28, Sednit, Pawn Storm, Group 74, Fancy Bear, Grizzy Steppe, APT29, Dukes, Group 100, Cozy Duke, Cozy Bear, Cozer |
| North Korea | Lazarus Group, Labyrinth Chollima, Bureau 121, Whois Hacking Team, Hidden Cobra, DarkHotel, Luder, Karba, APT-C-06, Dubnium, Fallout Team, Tapaoux |
| Iran | Cutting Kitten, TG-2889, Ghambar, COBALT GYPSY, Magic Hound, Timberworm, Elfin, Refined Kitter, APT33, Holmium, Shamoon2.0 |

We proposed an offensive cybersecurity framework that systematically organizes techniques used in cyber-attacks and defined offensive cybersecurity taxonomy based on this framework. Then, we described the intention and techniques of cyber-attacks on each offensive cybersecurity module such as encryption, network, web, malicious code and system. We chose fileless cyber-attacks and APT for cyberattack scoring. Fileless cyber-attacks discovered from 2014 to 2018 and APT cyber-attacks assumed to be supported by China, Russia, North Korea, and Iran—known as state-sponsored—were selected. Then, for the selected target, the techniques used in the offensive cybersecurity element were identified. In the case of malicious code, the Cyber Kill Chain (CKC) concept is applied because more detailed attack steps are used. The scoring score was calculated in two steps: (1) the first was to calculate the score for how many Offensive Cybersecurity elements were used in each stage of the CKC; (2) Second, we calculated how many cyber-attack techniques were used in 12 ATT&CK. Finally, the first and second steps were combined to calculate the final score. We utilized published analytical reports for investigating the techniques used.

The main contributions of the proposed scoring model using an offensive cybersecurity framework can be summarized as follows:

1.  We defined and derived the comprehensive offensive cybersecurity framework and taxonomy;
2.  We performed a content analysis of public reports of cyber-attacks and identified detailed techniques used in cyber-attacks;
3.  We provided a systematic scoring model based on the offensive cybersecurity framework;
4.  We calculated the score results of ten fileless and eight APT group cyber-attacks.

The remainder of this paper is structured as follows. Section 2 provides the background and presents a literature review of research pertaining to offensive cyber security. Section 3 explains our overall methodology toward offensive cyber security and addresses each element of offensive cyber security in detail. Section 4 discusses the scoring results assigned to cyber-attacks. Finally, Section 5 provides considerations for future work and conclusions.

## 2. Background and Literature Review

This section presents the literature review used in our proposed cyber-attack scoring model including security for CPS, offensive cybersecurity, and state-sponsored cyber-attacks.

### 2.1. Security for CPS

Khatoun and Zeadally [6] address concepts, architecture, and research opportunities of smart cities. They consider key components of smart cities to be smart living, smart mobility, smart economy, smart government, and smart people with the core technologies, Internet of Things (IoT), Internet of Data (IoD) and Internet of People (IoP). The major contribution of their paper is an analysis of the detailed elements of smart cities.

Miller and Valasec performed a car attack demonstration that greatly contributed to encouraging research into attacks and critical security vulnerabilities of autonomous cars in smart mobility [1,7,8]. Kim et al. [9] surveyed over 150 papers related to attacks on, and the defense of, autonomous vehicle. Adel et al. [10] summarized the cybersecurity challenges in smart cities in terms of their safety, security and privacy. Their survey showed that the data generated by smart cities emerge from people, homes, transportation, workplaces, schools, commerce and social activities. In addition, their analysis showed that the data in a smart city are not only linear, but have a circular structure that collects and uses the collected data repeatedly. Privacy is an essential aspect of the structure of the smart city. Kim [4] analyzed threats of smart homes based on the STRIDE threat model and constructed a systematic attack tree on the basis of their findings. The authors analyzed smart home communication techniques that usually connect the different nodes with Wi-Fi, ZigBee and Z-Wave. The authors purchased an established Smart Home in which Smart Home techniques had been adopted, and found various vulnerabilities in smart home equipment.

### 2.2. Offensive Cybersecurity

Dino Dai Zovi addressed the modern history of offensive cyber security [11]. He analysed the history of three generations of offensive cyber security research from 1993 to 2017. The first generation of offensive security was mainly the ventures of underground hackers from 1993 to 1997. During this period, major system vulnerabilities, such as buffer overflow, were investigated. The second generation, from 1997 to 2007, established security companies to provide security consulting and solutions. The third generation of offensive security is that of governments hiring experts who have offensive capability, or academia studying offensive cybersecurity. The Defense Advanced Research Projects Agency (DARPA) hosted the cyber grand challenge (CGC) to find vulnerabilities using automatic machines and artificial intelligence techniques. The USENIX Workshop of Offensive Technology focused on offensive cybersecurity technology from 2007. The movement to promote the necessity of offensive cyber security research has become an inevitable wave.

Richard et al. [12] analyzed the taxonomies of cyber security attacks, which are the first examined known attack type, and they examined actual attack cases based on their classification. The paper classified attack components into seven categories: Attacks, Reconnaissance, Vulnerability, Threats, Exploits, Payloads, and Effects. However, these seven components have substantial mutual conceptual overlap. In addition, the relationship between exploitation and payload is unclear because a payload is the core code part of the exploit code.

Simon et al. [13] classified network and computer attacks. In their paper, the categories in the first dimension of network and computer attacks are viruses, worms, buffer over-flows, denial of service attacks, network attacks, physical attacks, password attacks, and information gathering attacks. Then, 15 attacks were classified by adding the second, third, and fourth dimensions. Although they compiled a detailed taxonomy, because their work was published 10 years ago, recent cyber-attack techniques such as web attacks, mobile attacks, and IoT attacks are reflected to a limited extent.

Ben [14] noted that "sophisticated cyber-attacks" have increased dramatically over the last decade. The report identified various institutions, such as financial institutions, telecommunications agencies and state agencies, that have been attacked by "sophisticated cyber-attacks" and the meaning of the terms is not defined; thus, the meaning of "sophistication" is examined more intensively. The authors presented a framework for analyzing sophisticated cyber-attacks that included case studies to improve the formulation of offensive and defensive balance strategies in cyber operations. However, their report is not based on attack elements of the offensive cyber capabilities, but rather on a framework focused on attack methods. In addition, only some attack cases, such as the Stuxnet, were studied. Their study, therefore, does not follow a comprehensive and technical approach.

### 2.3. State-Sponsored Cyber-Attacks

Edward Snowden, the National Security Agency (NSA) contractor, unveiled an NSA hacking tool to the British newspaper *The Guardian* in 2013. This event drew the world's attention to state-sponsored hackers [15]. In general, Snowden's exposure revealed shocking tactics in the field of cybersecurity, as state-sponsored cyber-attacks were not readily visible to the public. In 2016, the Shadow brokers auctioned another code considered to be an NSA hacking tool [16]. The auction was unsuccessful, but later the NSA hacking tools were released to the world. The code released at the time, known as EternalBlue, exploited the Server Message Block (SMB) vulnerability that was used to create the Wan-naCry ransomware. Microsoft patched the SMB vulnerability under the name MS17-010 in 2017, necessitated by a widespread attack that was not reported by a whistle-blower in 2017. The United States also reportedly developed Stuxnet in 2010 with Israel with the aim of using cyber-attacks to delay Iran's nuclear development [17].

Another global operation by state-sponsored hackers was the involvement of Russian-sponsored hackers in the US presidential election. Bloomberg news said Russia's cyber-attack during the US presidential election in 2016 was very comprehensive and more powerful than ever before [18]. State-sponsored hacker organizations known to be sponsored by Russia are APT28 and APT29. The latter, also known as the Cozy Bear, is known to be sponsored by the Russian Federal Security Agency (FSB), a key Russian cyber security agency. APT28 is known as the Fancy Bear and is sponsored by the Main Intelligence Directorate (GRU), Russia's secret intelligence agency [19]. Benjamin et al. [20] described how Russian cyber actors disrupt and spy on the digital domain. Cyber tactics form part of the twenty-first century war strategy, and APT28 specifically attacked the Caucasus region and the North Atlantic Treaty Organization (NATO). They also noted that, in the 2016 US presidential election, Russia used psychological espionage to create a psychological impact on American society. Ben and Michael [21] stated that Russia's cyber operations are taking place across a very broad area of the United States. Consequently, the US government then took the Russian cyber-attack seriously and stressed that America should actively prepare for the network.

Various research reports on China's cyber operations are being published. Mandiant [22] researchers published that the PLA unit 61398, a Chinese military organization, has been continuously conducting cyber-attacks on the US government and civilian organizations. These attacks, unlike the one-off attacks by traditional attackers, have been designated APT because they have been continuously occurring for an extensive period of time. Therefore, the People's Liberation Army (PLA) Unit 61398 is named APT1. Antoine et al. [23] surveyed hacker organizations sponsored by China: APT16, APT17 (Aurora Panda), Shell_Crew, APT3

(Gothic Panda), APT15 (Ke3chang), APT12 (IXESHE), APT2 (Putter Panda), and APT30 (Naikon). The main characteristic of China's cyber operation is that it is designed to import industrial secrets from overseas advanced companies, mainly the exploitation of information. Kong et al. [24] investigated the link between North Korean cyber units and cyber operations on the North Korea cyber-attack capabilities. The authors predicted that cyber-attacks could not be expected to occur in only one country, anticipating linkages between political allies. According to a survey paper by Antoine et al. [23], Iran's nuclear weapons program was attacked by a joint operation between the US and Israel, known as the Olympic Games. Iran had been developing a malware named Shamoon two years after the Stuxnet attack and attacked financial and energy companies in the United States and Israel.

State-sponsored hacker groups have been investigated extensively [19]. In particular, after the involvement of Russian hackers in the US presidential election in 2016, additional research has been conducted. The attack techniques used by hacker groups supported by these countries are becoming more sophisticated; however, the level of these sophisticated cyber-attacks has not been measured.

## 3. Offensive Cybersecurity Framework

We systematized elements of the offensive cybersecurity framework to analyze the purpose and flow of cyber-attacks. Threat actors, which are individual hackers, cybercrime organizations, and nation-state hackers, are conducting cyber-attacks to achieve their goals such as financial gain or system destruction, as shown in Figure 1. Our framework is designed based on the threat actors, internet/network, and targets. The threat actors of cyber-attacks include Individual, Nation-State, and Cybercrime organizations. The Internet and Network are used in the process of accessing the attack target, and Public Network, Proxy, Virtual private network (VPN), and Darknet/Deepweb are used for this. The target of an attack is largely divided into Organizations, and CPS. Within the Organization, there are Web Servers and Web Application Servers (WAS) that are open to the outside, and there are personal computers and mobile phones inside the organization and documents and information in them. Organizations also have internal systems and databases containing important information. Recent attack targets include CPS and Persons. The detailed attack targets of the CPS include Smart Homes, Smart Mobility, Smart Economy, and Smart People. Offensive cybersecurity refers to the attack categories that cyber attackers need to compromise, including encryption, networks, web, malware, and systems.

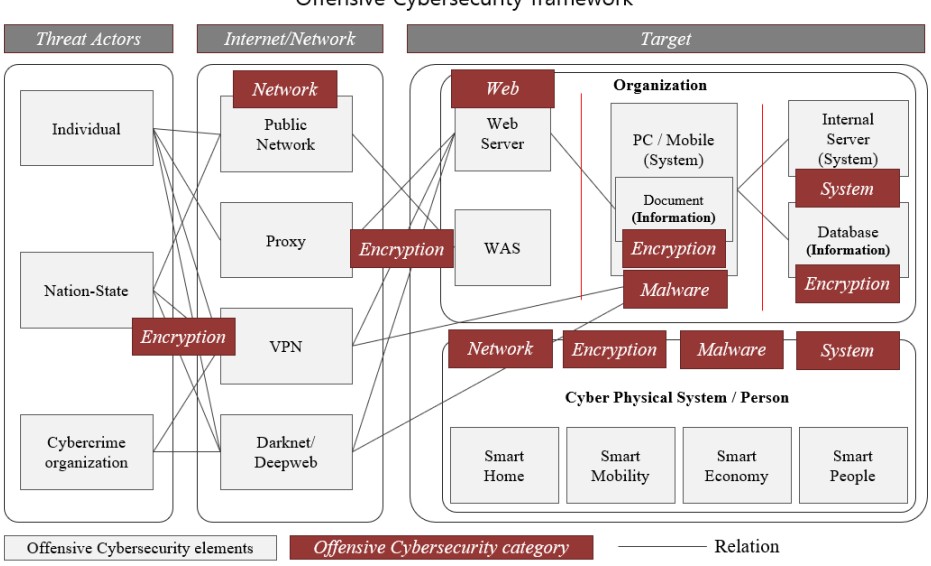

**Figure 1.** Offensive cybersecurity framework.

Figure 2 shows the taxonomy of Offensive Cybersecurity in detail. Encryption is a method used by cyber attackers to encrypt information, and there are Substitution, Symmetric, Asymmetric, and Hash Algorithms. Examples of network-related cyber-attacks include sniffing, spoofing, and Denial of Service (DoS) and Distributed Denial of Service (DDoS). Web attacks include Injection, Webshell upload, Authentication Access Control, File Download, and Cross-Site Scripting (CSS). Malicious code is more complex, and there is generally key logging, Remote Administration Tools (RAT), DLL injections, script attacks, and so forth, and Obfuscation, Packing, Cryptor and Protector protect the malicious code itself. Lastly, the system has detailed attack targets such as Applications, Services, Operating System (OS)/Kernel and Hypervisor, memory corruption attacks, such as Buffer Overflow (BOF) and Heap Overflow (HOF), and techniques to bypass security functions such as Return-oriented programming (ROP). We analyzed each offensive cybersecurity element to identify detailed techniques used by cyber-attackers.

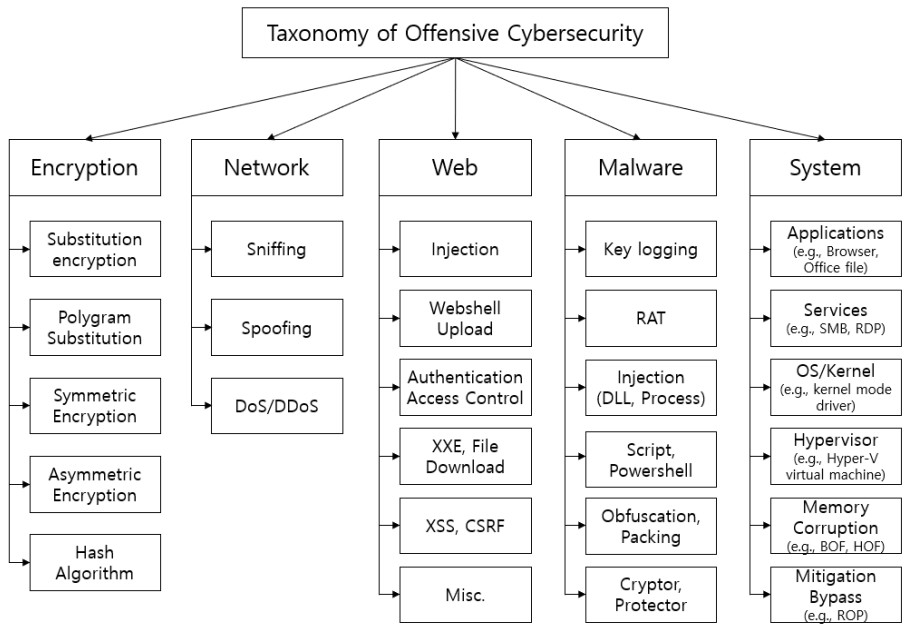

**Figure 2.** Offensive Cybersecurity Taxonomy.

### 3.1. Encryption

Encryption is the crucial technique used in cyber-attacks for various purposes. Traditional encryption techniques are used to encrypt confidential information to protect adversary users. The first encryption method was a substitution encryption. Representative encryption methods are the Caesar Cipher (B.C. 500) and the Monoalphabetic Cipher. Then, Polygram substitution, such as the Vigenere cipher (1585~1863) and Enigma cipher (1930), were developed. The creation of modern computing systems was followed by the development of symmetric encryption methods such as the Data Encryption Standard (DES) and Advanced Encryption Standard (AES). These two standards use the same key for encryption and decryption. As a result, an asymmetric encryption method was developed to share the secret key securely. Representative techniques in an asymmetric encryption are the Diffie–Hellman Key Exchange, Rivest–Shamir–Adleman (RSA) and the elliptic-curve cryptography (ECC) method. In the case of developed websites, login credentials, such as passwords, need to be saved in secure databases. In view of this circumstance, hash algorithms (MD5, SHA1 and SHA2) are used to protect users' credentials in the database.

Ransomware uses a symmetric encryption method to encrypt users' valuable files such as images and documents. Ransomware also uses an asymmetric encryption algorithm with the attacker's private key to protect the encryption key that is used in a symmetric encryption [25]. In the era in which state-sponsored hackers are becoming increasingly active, these more elegant attackers are using various encryption methods to hide traces

of their activities. After gaining confidential information from a victim, they use various encryption methods to evade being monitored by the victim's system and network. APT28 (sponsored by Russia) uses encrypted POST data to send a command to command and control (C&C) server with an obfuscation base64 (block cipher). The encryption schemes used from an offensive cybersecurity perspective are listed in Table 2, and the cases in which these methods were used in cyber-attacks are compared.

**Table 2.** Cryptography techniques and cyber-attack cases.

| Category | Encryption Method | Cyber-Attack Case |
|---|---|---|
| Substantial Encryption | Caesar cipher<br>Rotation 13 (ROT13) | Coin Locker<br>CryptoShield |
| Symmetric Encryption | DES<br>3DES<br>AES | -<br>JobCrypter Ransomware<br>Ransomware [25], APT29 |
| Asymmetric Encryption | RSA<br>ECC | Ransomware, APT28, APT29<br>OphionLocker Ransomware |
| Block Encryption | Rivest Cipher (RC)4<br><br>RC5<br>RC6<br>Base64 | ProjectSauron [26],<br>DarkHotel [27]<br>ProjectSauron<br>ProjectSauron<br>APT28, APT29 |
| Hash | Message-Digest (MD)5<br>Secure Hash Algorithm (SHA) | -<br>NetWalker ransomware |
| Custom | Custom | DarkUniverse |

### 3.2. Network

Most well-known cyber-attacks techniques related to networks are sniffing, spoofing, and Denial of Service. Sniffing is known as a passive network attack because it does not directly attack the target's computer. Spoofing, on the other hand, is an attack that deceives the network protocol and includes Internet Protocol (IP), Domain Name System (DNS), and Address Resolution Protocol (ARP) spoofing. Spoofing is classified as a representative active network attack.

As the Internet evolves, many organizations offer their products and services over the Internet. Network availability has become very important in this process, particularly in the case of CPS. Attackers are using networks to perform denial of service (DoS) attacks and distributed denial of service (DDoS) attacks as a way to attack network availability. DoS and DDoS attacks include Ping of Death, Synchronize (SYN) flooding, and Hypertext Transfer Protocol (HTTP) Get flooding techniques. From the point of view of offensive security, the main objective of a network attack is to conceal the identity and location of the attacker. IP addresses are typically used to identify devices on the network. Attackers hide their source through a proxy or Tor browser to prevent their IP addresses from being revealed.

Hassan et al. [28] and Nazrul et al. [29] classified network cyber-attacks as follows: Information gathering, DoS and DDoS attacks, spoofing, TCP session hijacking, probe, application layer, malformed packet, amplification, and protocol exploit attacks. Dileep et al. [30] classified network attacks based on the network layer. We adopt this concept to construct our offensive framework.

### 3.3. Web

From the perspective of offensive cybersecurity, the Web is a prime target. Most companies promote their products and services on websites; however, it also makes it easy for attackers to conduct cyber-attacks through a website. Because of this, web hacking frequently occurs and related attack techniques are constantly being studied. We

investigated web attack techniques and adopted the Open Web Application Security Project (OWASP) for our methodology. The OWASP organization was formed in 2001 and reports ten dangerous vulnerabilities on its website every three years. The web vulnerability that was consistently ranked number one from 2010 to 2017 is an injection vulnerability. This can be used to fetch database information from a web site or to gain system privileges on the web server. A typical attack among those intended to hack websites, along with injection attacks, is a defacement attack. It is used to steal information, whereas website tampering attacks reveal and hack. Typically, hacktivist organizations hack websites because of their political orientation. Among the attacks used to alter web pages, a typical hacktivist attack uploads a malicious script, known as a web shell, to a web server to alter the web page. We determined that a sizeable proportion of cyber-attacks use compromised websites to spread their malicious code. Table 3 shows the top ten most dangerous web vulnerabilities from 2010 to 2017.

**Table 3.** OWASP Top 10 (From 2010 to 2017).

| Top 10 | 2010 | 2013 | 2017 |
|---|---|---|---|
| A1 | Injection | Injection | Injection |
| A2 | Cross-Site Scripting (XSS) | Broken Authentication and Session Management | Broken Authentication |
| A3 | Broken Authentication and Session Management | Cross-Site Scripting (XSS) | Sensitive Data Exposure |
| A4 | Insecure Direct Object References | Insecure Direct Object References | XML External Entities (XXE) |
| A5 | Cross-Site Request Forgery (CSRF) | Security Misconfiguration | Broken Access Control |
| A6 | Security Misconfiguration | Sensitive Data Exposure | Security Misconfiguration |
| A7 | Insecure Cryptographic Storage | Missing Function Level Access Control | Cross-Site Scripting (XSS) |
| A8 | Failure to Restrict URL Access | Cross-Site Request Forgery (CSRF) | Insecure Deserialization |
| A9 | Insufficient Transport Layer Protection | Using Known Vulnerable Components | Using Components with Known Vulnerabilities |
| A10 | Unvalidated Redirects and Forwards | Unvalidated Redirects and Forwards | Insufficient Logging and Monitoring |

*3.4. Malware*

A considerable amount of research has been devoted to uncovering malware techniques. Representative research is the form of ATT&CK. Kris and Christian [31] explained malware techniques based on the ATT&CK step. Techniques used during the malware execution step include: "Execution through API (CreateProcessA function)", "using Rundll32", "Command-Line Interface (cmd.exe)", "Service Execution (register or execute as a service)", "PowerShell", and "Windows Management Instrumentation (WMI)". Persistence techniques are "Registry Run Key", "New Service", "Modify Existing Service", "Hooking", "Schedule Task", and "Image File Execution Options Injection". Techniques for Privilege Escalation include Process Injection and Access Token Manipulation. Ekta et al. [32] proposed a malware threat assessment using a fuzzy logic paradigm.

Paul et al. [33] described the behavior of malware as: persistence, configuration, process injection, information stealing and injection, network communications, backconnect, screenshot and video capture, and anti-analysis. Anti-analysis uses obfuscation, packing, cryptor, and protector techniques to confuse analysis malware. In the same way that bullets are important in war, malware is also crucial ammunition in cyber-attacks. Malware uses a wide variety of technologies, and unlike other elements, the stage at which the malicious code is executed is very complicated. According to Paul et al. [33], obfuscation and packing is a technique for hiding malware, and recently, the DLL Side-loading technique has been

utilized. We constructed the technology and stages used by malicious code based on the MITRE ATT&CK. At the ATT&CK stage, the technology focused on malicious codes was quantified, and Execution, Persistence, Privileges Escalation, Defense Evasion, Credential Access, Lateral Movement, Command and Control, Exfiltration, and Impact were selected as representative technologies as listed in Table 4.

**Table 4.** Malicious code techniques.

| Stages | Techniques |
| --- | --- |
| Execution | Script, Powershell, Cli, Schedule task, Signed binary, User execution, Service execution, Rundll32, Mshta, WMI |
| Persistence | Registry run keys, Scheduled task, New service |
| Privilege Escalation | Process injection, Exploit, Access token manipulation, New service |
| Defense Evasion | Hidden files, Modify registry, Permission modify, Process injection, Packing, Deletion, Obfuscate, Masqurade, deobfuscate, Disable tools, Mshta, Indicator rm |
| Credential Access | Credential dumping, Brute force, Credential in files, Pass the Hash |
| Lateral Movement | WA Share, Exploit remote, Remote file copy |
| Command and Control | Common ports, Multi-hop proxy, Multilayer encryption, Remote file copy, Uncommon ports, Data encoding, Data obfuscation |
| Exfiltration | Automated exf. Exf. over alt. protocol, Data encrypted, Exf. over C&C |
| Impact | Disk struct wipe, Encrypt data, Inhibit recovery, Service stop |

*3.5. System*

System attack has been the subject of extensive research. Systems consist of many layers: Application, Services, OS and Kernel, and Hypervisor. The prime vulnerability of systems and applications is a memory corruption. Mitigation techniques have been steadily researched; in addition, mitigation bypass techniques have also been developed continuously. A system is divided into four layers: Applications, Services, OS and Kernel, and Hypervisor for the cloud. Application categories include browsers, Microsoft Office, and Adobe programs. Services represent specific functions that are provided from outside the system and include the SMB and the remote desktop protocol (RDP). The operating system and kernel level are other prevalent attack targets. In the cloud environment, the hypervisor is the basis on which the operating system is run and also a critical target of offensive cyber-attacks. Most types of system vulnerability are in the memory corruption category. These techniques are buffer overflow (BOF), heap overflow (HOF), and integer overflow. A number of mitigation techniques have been developed to defend against system vulnerability. Data execution prevention (DEP) in Windows and the no-execute (NX) bit are designed to defend the execute shell code in the stack area. Address space layer randomization (ASLR), which is also adopted as a defense against memory corruption attacks, changes the stack address after each execution. Offensive techniques to bypass these mitigations are steadily being researched; specifically, return-oriented programming (ROP) is a major attack technique used to bypass a stack address defense.

## 4. Cyber-Attacks Evaluation

We evaluated each cyber-attack case by modeling offensive cybersecurity. We adopted the proposed methodology by selecting numerous fileless and APT cyber-attack cases. The reason we select fileless and APT is that these kinds of cyber-attacks have advanced the sophistication of cyber-attack techniques.

*4.1. Dataset*

We used datasets that contain cases of two types of cyber-attacks: fileless cyber-attacks and APT group cyber-attacks. We chose ten recent fileless cyber-attacks listed in Table 5 from the dataset to evaluate our scoring model. The ten selected fileless cyber-attacks were Poweliks, Pozena, Duqu 2.0, Kovter, Petya, Sorebrect, WannaCry, Magniber, Emotet, and GandCrab.

**Table 5.** Fileless cyber-attack dataset.

| No | Cyber-Attack | Year | SHA256 |
|----|--------------|------|--------|
| 1 | Poweliks | 2014 | - |
| 2 | Rozena | 2015 | c23d6700e93903d05079ca1ea4c1e36151cdba4c5518750dc604829c0d7b80a7 |
| 3 | Duqu 2.0 | 2015 | 52fe506928b0262f10de31e783af8540b6a0b232b15749d647847488acd0e17a |
| 4 | Kovter | 2016 | - |
| 5 | Petya | 2017 | 027cc450ef5f8c5f653329641ec1fed91f694e0d229928963b30f6b0d7d3a745 |
| 6 | Sorebrect | 2017 | 4142ff4667f5b9986888bdcb2a727db6a767f78fe1d5d4ae3346365a1d70eb76 |
| 7 | WannaCry | 2017 | ed01ebfbc9eb5bbea545af4d01bf5f1071661840480439c6e5babe8e080e41aa |
| 8 | Magniber | 2017 | c21887eaa1e31b9220d0807d3a7d0f30421ab6f80cfc1c556d534587dd9e6343 |
| 9 | Emotet | 2017 | 70903a9ef495edd8de01a61f8e9862a037b0dee327d7f92f93ef69e33e461764 |
| 10 | GandCrab | 2018 | 643f8043c0b0f89cedbfc3177ab7cfe99a8e2c7fe16691f3d54fb18bc14b8f45 |

We also selected APT Group and Operation from the APT group list [5]. In this study, we first investigated six nations: China, Russia, North Korea, Iran, Israel, and nations in the Middle East. These countries and regions have 87, 20, 9, 9, 2 and 17 cyber-attack groups, respectively, as listed in Table 6. We selected the APT groups of China, Russia, North Korea, and Iran because they are known publicly. Even though China has the most cyber-attack groups, the mere number of cyber-attack groups does not indicate a nation's cyber-attack capabilities. For example, in comparison to China, Israel has only two cyber-attack groups. China has a smattering of small cyber-attack groups, however Israel's Unit 8200 group is known to be the most powerful group.

**Table 6.** Cyber-attack group dataset.

| Nations | Counts | APT Groups Common Name |
|---------|--------|------------------------|
| China | 87 | Comment Crew, APT2, UPS, IXESHE, APT16, Hidden Lynx, Wekby, Axiom, Winnti Group, Shell Crew, Naikon, Lotus Blossom, APT6, APT26, Mirage, NetTraveler, Ice Fog, Beijing Group, APT22, Suckfly, APT4, Pitty Tiger, Scarlet Mimic, C0d0so, SVCMONDR, Wisp Team, Mana Team, TEMP.Zhenbao, SPIVY, Mofang, DragonOK, Group 27, Tonto Team, TA459, Tick, Lucky Cat, APT40, PassCV, BARIUM, LEAD, Iron Group, Anchor Panda, Big Panda, Electric Panda, Eloquent Panda, Emissary Panda, Foxy Panda, Gibberish Panda, Goblin Panda, Hammer Panda, Hurricane Panda, Impersonating Panda, Judgement Panda, Karma Panda, Keyhole Panda, Kryptonite Panda, Mustang Panda, Night Dragon, Nightshade Panda, Nomad Panda, Pale Panda, Pirate Panda, Poisonous Panda, Predator Panda, Radio Panda, Sabre Panda, Spicy Panda, Stone Panda, Temper Panda, Test Panda, Toxie Panda, Union Panda, Violin Panda, Wet Panda, Calypso, Tropic Trooper, APT41, Poison Carp, AVIVORE, APT-C-01, DarkUniverse, Taskmasters, GALLIUM, RANCOR, ChinaZ, APT-C-37, APT-C-27 |
| Russia | 20 | Sofacy, APT29, Turla Group, Energetic Bear, Sandworm, FIN7, FIN8, Inception Framework, TeamSpy Crew, BuhTrap, Carberb, FSB 16th & 18th Centers, Cyber Berkut, WhiteBear, GRU GTsST (Main Center for Special Technology), VOODOO BEAR, TEMP.Veles, Zebrocy, SectorJ04, FullofDeep |
| North Korea | 9 | Lazarus Group, Group13, DarkHotel, Andariel, Kimsuki, NoName, OnionDog, TEMP.Hermit, Stardust Chollima |
| Iran | 9 | Cutting Kitten, Shamoon, Clever Kitten, Madi, Cyber fighters of Izz Ad-Din Al Qassam, Chafer, Prince of Persia, Sima, Oilrig |
| Israel | 2 | Unit 8200, SunFlower |
| Middel East | 17 | Molerats, AridViper, Volatile Cedar, Syrian Electronic Army (SEA), Cyber Caliphate Army (CCA), Ghost Jackal, Corsair Jackal, Extreme Jackal, Electric Powder, APT-C-23, APT-C-27, Dark Caracal, Tempting Cedar, Sandcat, Group WITRE, ZooPark, APT-C-37 |
| Total | 144 | |

*4.2. Investigation*

We calculated the cyber-attack score based on the Open Source Intelligence (OSINT) method. Many cybersecurity companies publish an analysis of cyber-attack cases, and certain countries, such as the United States and South Korea, publish reports with analyses of cyber-attack cases. We analyzed these reports to identify the techniques that were being used by the cyber-attack groups.

To identify detailed techniques that were used for each cyber-attack, we selected representative examples of a cyber-attack for each country as well as fileless cyber-attacks. In the first step of our analysis, we used MITRE ATT&CK cyber-attack group artifacts to identify the cyber-attack techniques that were used. Next, we analyzed the cyber-attack techniques in detail based on our proposed offensive cybersecurity framework for each representative cyber-attack as listed in Tables 7 and 8.

**Table 7.** Techniques used in the fileless cyber-attacks samples.

| Fileless Cyber-Attack | Techniques |
|---|---|
| Poweliks | MS Office Macro vulnerability<br>Inject malicious script into the registry<br>Execution of registry value using Rundll32.exe<br>Execution of registry value encoded using Jscript.Encode<br>Use of Powershell script encoded with Base64<br>Verification of registry key and path of executed files<br>DLL execution through Powershell schript (injection using dllhost.exe)<br>Deleting files after every operation<br>Dllhost.exe resides<br>Send a user's system information to C&C server through TCP communication |
| Kovter | Social engineering techniques using email attachments<br>Inject malicious script into the registry<br>Execution of registry value using Mshta.exe<br>Execution of registry value encoded using Jscript.Encode<br>Use of Powershell script encoded with Base64<br>Injecting codes through Powershell script<br>Deleting files after every operation<br>Regsvr32.exe resides<br>Send a user's system information to C&C server through TCP communication |

**Table 8.** Techniques used in the APT Group cyber-attacks.

| Cyber-Attack | Techniques Used |
|---|---|
| APT29 | Network: Multi-hop Proxy, Commonly Used Port, Domain Fronting, Spearphishing Attachment, Spearphishing Link, Standard Non-Application Layer Protocol, User Execution<br>Encryption: Obfuscated Files or Information, Software Packing<br>System: Accessibility Features, Bypass User Account Control, Exploitation for Client Execution, Pass the Ticket<br>Malware: File Deletion, Indicator Removal on Host, PowerShell, Registry Run Keys/Startup Folder, Rundll32, Scheduled Task, Scripting, Shortcut Modification, Windows Management Instrumentation, Windows Management Instrumentation Event Subscription<br>Used tools: CloudDuke, CosmicDuke, CozyCar, GeminiDuke, HAMMERTOSS, meek, Mimikatz, MiniDuke, OnionDuke, PinchDuke, POSHSPY, PowerDuke, PsExec, SDelete, SeaDuke, Tor |

**Table 8.** *Cont.*

| Cyber-Attack | Techniques Used |
|---|---|
| Lazarus Group | Encryption: Custom Cryptographic Protocol, Data Compressed, Data Encoding, Data Encrypted, Obfuscated Files or Information, Standard Cryptographic Protocol<br>Network: Commonly Used Port, Connection Proxy, Exfiltration Over Alternative Protocol, Exfiltration Over Command and Control Channel, Fallback Channels, Multiband Communication, Spearphishing Attachment, Standard Application Layer Protocol, Uncommonly Used Port<br>Web: Drive-by Compromise<br>Malware: Access Token Manipulation, Account Manipulation, Application Window Discovery, Bootkit, Brute Force, Command-Line Interface, Credential Dumping, Data from Local System, Data Staged, Disabling Security Tools, Disk Content Wipe, Disk Structure Wipe, File and Directory Discovery, File Deletion, Hidden Files and Directories, Input Capture, New Service, Process Discovery, Process Injection, Query Registry, Registry Run Keys/Startup Folder, Remote File Copy, Resource Hijacking, Scripting, Service Stop, Shortcut Modification, System Information Discovery, System Network Configuration Discovery, System Owner/User Discovery, System Shutdown/Reboot, System Time Discovery, Timestomp, Windows Admin Shares, Windows Management Instrumentation<br>System: Compiled HTML File, Exploitation for Client Execution, Remote Desktop Protocol, User Execution<br>Used tools: AuditCred, BADCALL, Bankshot, FALLCHILL, HARDRAIN, HOPLIGHT, KEYMARBLE, Mimikatz, netsh, Proxysvc, RATANKBA, RawDisk, TYPEFRAME, Volgmer, WannaCry |

### 4.3. Scoring Procedure

We collected almost 150 reports with analyses of attacks and websites using the links that had already been collected by the APT Group list [5]. Then, we searched each paper for cyber-attack techniques using the keyword-base addressed in Section 3 (Offensive Cybersecurity elements). Finally, we calculated the cyber-attack score based on the offensive cybersecurity elements. Figure 3 and Algorithm 1 shows the overall scoring procedure. Additionally, one cyber-attack group can perform various cyber-attack operations. In this case, we selected representative cyber-attack operations for each cyber-attack group. We used the Lockheed Martin Cyber Kill-Chain process to identify the cyber-attack techniques for each step [34].

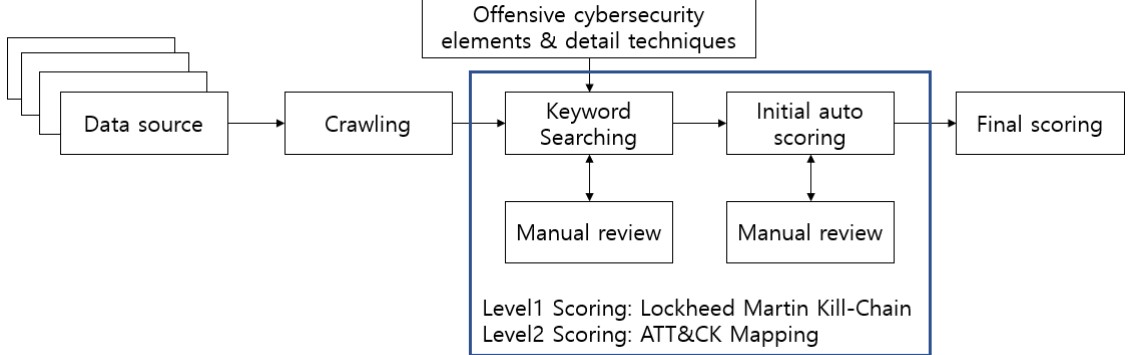

**Figure 3.** Overall scoring procedure.

MITRE ATT&CK evaluations showed the scoring result for some APT groups such as APT3, APT29, and Carbanak+FIN7. This evaluation was based on the 20 attack stages; however, it only focused on the malware itself rather than mapping the overall chain of the cyber-attack. Thus, our scoring approach differs from that used in the ATT&CK evaluation.

In addition, our approach provides a more comprehensive description of cyber-attack techniques to arrive at a cyber-attack score.

---

**Algorithm 1** Generating score for each cyber-attack.

---

1: **Input :** Data source-malware analysis report
2: **Output :** Cyber-attack scoring
3: *N* ← number of cyber-attack cases
4: *Element[score]* ← 0
5: **for** *k* ← 1 to *N* **do**
6:    *keyword* ← *keyword file*
7:    **if** *content == keyword* **then**
8:       *score* ← *score* + 20
9:       *element*[*score*] ← *score*
10:    **end if**
11: **end for**
12: Return element[score]

---

We evaluated cyber-attacks at two levels to calculate the score. Level 1 uses the top offensive cybersecurity elements for each cyber-attack with cyber kill chain phases. We identified the offensive cybersecurity elements that were used in each cyber-attack case. For this purpose, we analyzed the cyber-attacks and mapped them with the Lockheed Martin Cyber Kill-Chain. The Cyber Kill-Chain has seven phases and a total of five cybersecurity high-level elements. We awarded 20 points to each cybersecurity element.

Combining the cyber-attack techniques used in each of the offensive cybersecurity model, the cyber-attack complexity is expressed as follows.

$$Z = Sum \sum_{k=0}^{n} \frac{UOCSM}{TOCSM}. \tag{1}$$

In Equation (1), UOCSM denotes 'Using Offensive Cybersecurity Modules', TOCSM means 'Total Offensive Cybersecurity Modules', and Z represents the utilization of offensive cybersecurity elements. Thus, each phase can earn a maximum of 100 points, which means that the maximum total score (TOCSM) is 700 points.

Level 2 utilized data used by ATT&CK techniques in each element for cyber-attacks. ATT&CK has 12 steps for conducting cyber-attacks. We calculated the sum of technologies used in each ATT&CK phase of the cyber-attack.

*4.4. Scoring*

4.4.1. Cyber-Attack Scoring Result with Cyber Kill Chain

We analyzed in detail the cyber-attack techniques for each fileless cyber-attack. Figure 4 presents an example of the scoring result for a Poweliks fileless cyber-attack. For Powerliks, in the initial Reconnaissance phase of Cyber Kill Chain, the attackers obtained e-mail addresses of post office workers in the US and Canada. Then, in the Weaponization phase, the attacker created an MS Word document file and inserted malicious code using the Macro vulnerability inside. In the delivery stage, an email containing a malicious program was disguised as a normal email and delivered to the employee's email address. In the Exploitation phase, malicious code was executed using the MS macro function vulnerability. It also inserted a malicious script into the registry and rendered the name of the registrar key unreadable. Additionally, the encoded registry value was executed using the JScript.Encode function. In the installation step, the Base64-encoded PowerShell script was executed. DLL injection was also performed using a PowerShell script, and the malicious DLL is packed with MPRESS. In addition, Poweliks registers malicious code in the automatic program startup registry to perform permanent attacks. In the Command and Control phase, TCP connections are transformed into the two IP addresses that are

estimated to be servers. In the Action on objectives step, information about the user's PC is collected and information is transmitted to the attacker's server.

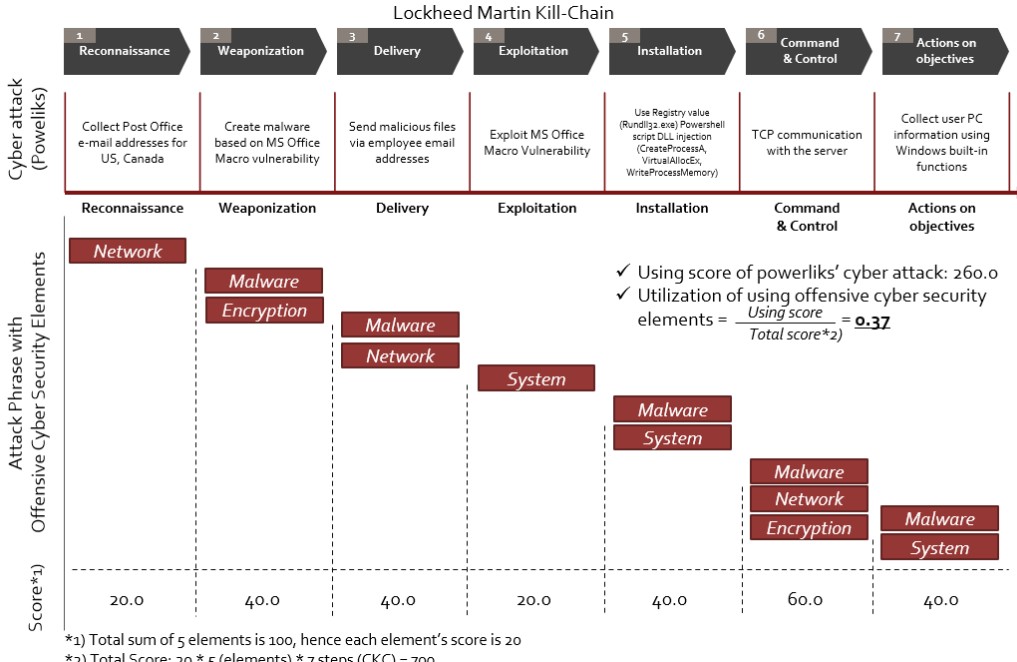

**Figure 4.** Example of scoring using a Poweliks cyber-attack.

For the WannaCry case, it does not appear to have a separate Reconnaissance stage. In the Weaponization phase, the attacker creates a malicious program disguised as the icon of a normal program. In the delivery phase, the attacker uploads a malicious file disguised as a normal file online, and then the victim downloads it. In the Exploitation phase, the MSSecsvc 2.0 service is installed on the victim's PC, and files hidden in the program's resources are dropped and executed. In the installation phase, the dropped program is registered in the registry run key, and it is automatically executed whenever the PC is booted. In the Command and Control phase, communication occurs via the Tor network, and port 9050 is left open should communication with an external server be required. Actions on objectives encrypt all data except for data in a specific file path. The volume shadow file is deleted using Vssadmin on the infected PC. The peculiarity is that the SMB vulnerability also enables the shellcode to be transmitted to a computer on a shared network, and the vulnerability of the PC results in the same process being used to infect the ransomware.

We selected the Lazrus cyber-attack for the APT group from four countries. Figure 5 shows an example of the scoring result for the Lazarus APT cyber-attack. In the Lazarus case, the cyber-attack collected a post-office e-mail address and investigated specific targets with network proving techniques in the reconnaissance step. In the Weaponization step, Lazarus developed malware by exploiting the 0-day vulnerability of Adobe software. Many encryption techniques were adopted during the development of malware. Web, malware, and network techniques were used in the delivery step. In the Exploitation step, Lazarus used various 0-day exploits; thus, we evaluated the system and malware element in the exploitation step. Malware, system, and encryption techniques were used in the Installation step, which used TCP port 443 with some payloads for the implementation of SSL encryption. Actions on the Objectives step in the cyber kill chain were performed by gaining system information, downloading and uploading files, and using the execution command.

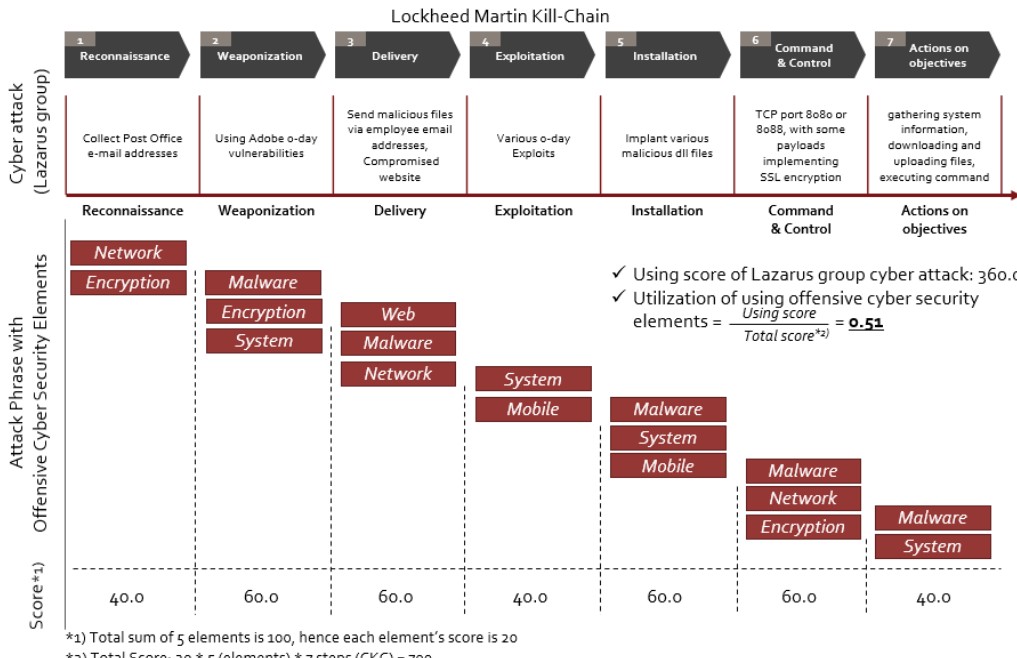

**Figure 5.** Cyber-attack Score example for Lazarus Group.

Through these methods, we calculated the scoring result of cyber-attacks with Cyber Kill Chain as listed in Table 9. The result using Cyber Kill Chain shows Poweliks (0.3714), Rozena (0.3429), Duqu 2.0 (0.3429), Kovter (0.3429), Petya (0.4286), Sorebrect (0.2857), WannaCry (0.3714), Magniber (0.2857), Emotet (0.3429), and GandCrab (0.3429) for fileless cyber-attacks. For APT cyber-attacks, it shows APT1 (0.4000), Emissary Panda (0.4857), APT29 (0.4286), SectorJ04 (0.4571), Lazarus Group (0.5143), APT38 (0.4286), Chafer (0.4000), MuddyWater (0.4286). The average score for fileless and APT cyber-attacks is 0.3459, 0.4318, respectively. This shows that APT cyber-attacks use more Cyber Kill Chain techniques than fileless cyber-attacks.

**Table 9.** Scoring result of cyber-attacks using Cyber Kill Chain.

| No. | Cyber-Attacks | UOCSM | Lockheed Martin Cyber Kill Chain Phase | | | | | | | Total |
|-----|---------------|-------|----|----|----|----|----|----|----|-------|
| | | | R | W | D | E | I | C | A | |
| 1 | Poweliks | 260.0 | 20 | 40 | 40 | 20 | 40 | 60 | 40 | 0.3714 |
| 2 | Rozena | 240.0 | 20 | 40 | 20 | 60 | 40 | 40 | 20 | 0.3429 |
| 3 | Duqu 2.0 | 240.0 | - | 40 | 40 | 40 | 40 | 40 | 40 | 0.3429 |
| 4 | Kovter | 240.0 | 20 | 40 | 40 | 40 | 40 | 20 | 40 | 0.3429 |
| 5 | Petya | 300.0 | 20 | 60 | 40 | 60 | 60 | - | 60 | 0.4286 |
| 6 | Sorebrect | 200.0 | - | 40 | 20 | 40 | - | 40 | 60 | 0.2857 |
| 7 | WannaCry | 260.0 | - | 40 | 40 | 40 | 40 | 40 | 60 | 0.3714 |
| 8 | Magniber | 200.0 | - | 40 | 20 | 40 | 40 | - | 60 | 0.2857 |
| 9 | Emotet | 240.0 | - | 40 | 40 | 40 | 40 | 40 | 40 | 0.3429 |
| 10 | GandCrab | 240.0 | - | 40 | 20 | 40 | 40 | 40 | 60 | 0.3429 |
| 11 | APT1 | 280.0 | 20 | 60 | 40 | 40 | 60 | - | 60 | 0.4000 |
| 12 | Emissary Panda | 340.0 | 40 | 40 | 60 | 40 | 60 | 60 | 40 | 0.4857 |
| 13 | APT29 | 300.0 | 40 | 20 | 40 | 40 | 60 | 60 | 40 | 0.4286 |
| 14 | SectorJ04 | 320.0 | 20 | 40 | 40 | 40 | 60 | 60 | 60 | 0.4571 |
| 15 | Lazarus Group | 360.0 | 40 | 60 | 60 | 40 | 60 | 60 | 40 | 0.5143 |
| 16 | APT38 | 300.0 | 40 | 40 | 60 | 40 | 40 | 40 | 40 | 0.4286 |
| 17 | Chafer | 280.0 | 40 | 40 | 40 | 40 | 40 | 40 | 40 | 0.4000 |
| 18 | MuddyWater | 300.0 | 20 | 40 | 40 | 40 | 60 | 60 | 40 | 0.4286 |

4.4.2. Cyber-Attack Scoring Result with ATT&CK

Owing to the nature of cyber-attacks, increasingly complicated techniques are included in malicious code. Thus, when calculating the cyber-attack score, we conducted a more in-depth analysis of the malicious code. An analyze process was carried out on the 12 stages of the MITRE ATT&CK, and the results are shown in Figure 6.

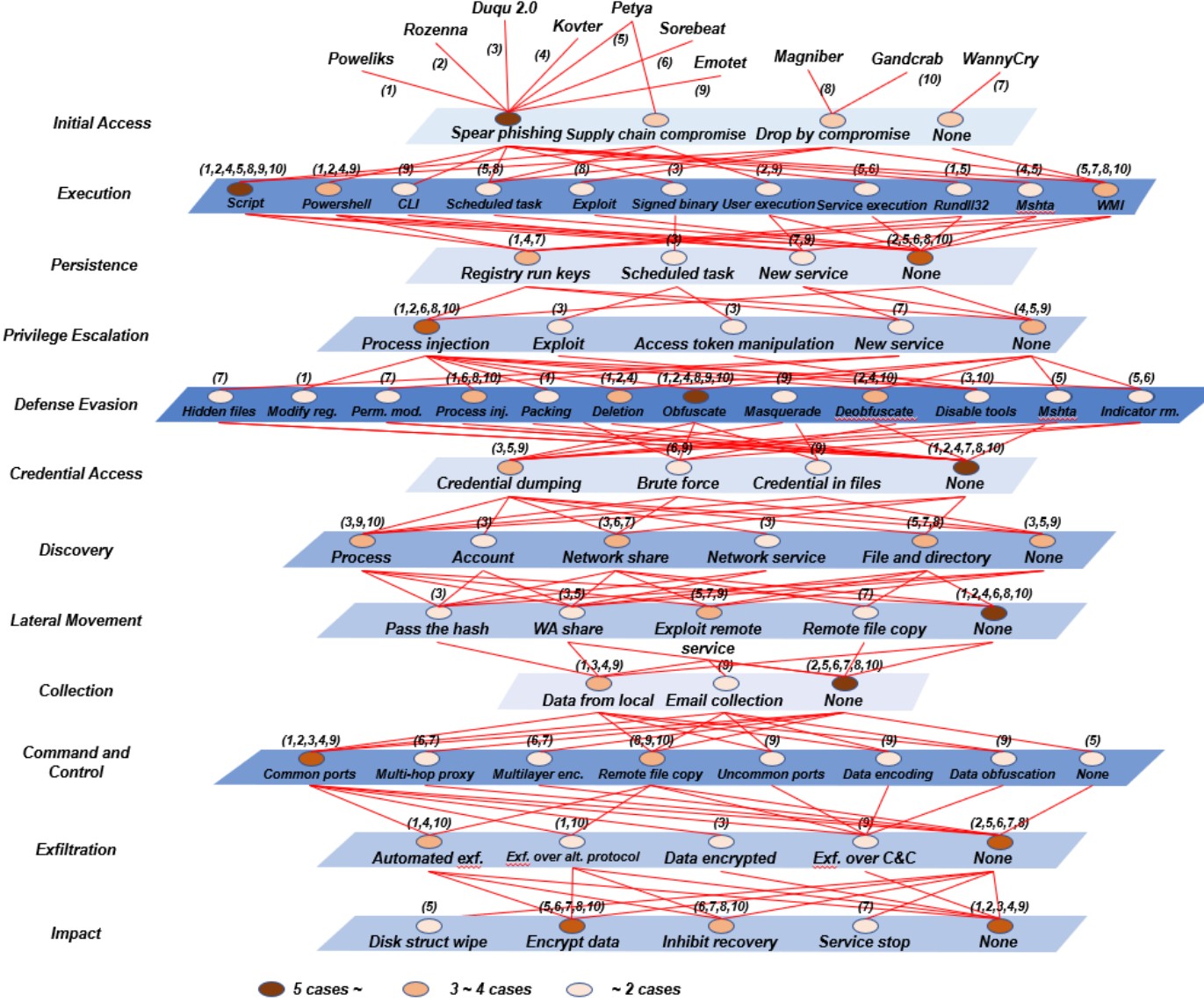

**Figure 6.** Attack techniques of fileless cyber-attack [35].

In this figure, each layer shows the stages of the cyber-attack in MITRE ATT&CK. The order of layers is the flow of an attack. The red line indicates the connection of a technique used by a malicious code with the previous techniques used. At each stage, the shade of blue plane indicates the cyber-attack techniques. In addition, the color of the circle indicates the number of cyber-attacks that use it. For instance, if the number of cyber-attack types is five or more, the circle's color is dark brown; if it is three or four, it is orange, and if it is two or less, it shows an apricot color.

For example, the result of Duqu 2.0 mapping to ATT&CK is as follows. The initial access step used a spearphishing attachment. Signed binary and proxy execution were used in the Execution step. In the Persistence step, the scheduled task technique was used, and in the Privilege escalation step, exploitation for privilege escalation and access

token manipulation techniques were used. In the Defense evasion step, disabling security tools were used, and in the Credential access step, credential dumping was used. In the Discovery step, process discovery, account discovery, network share discovery, and network service scanning were used. Data from local system technology was used in the Collection step, and a commonly used port was used in the Command and Control step. Data encrypted technology was used in the Exfiltration step.

Through these methods, we calculated the scoring result of cyber-attacks with the MITRE ATT&CK as listed in Table 10. The following is a description of the results for Petya, which earned a score of 0.2581. In the initial access step, a spear phishing attachment and supply chain compromise technologies were used. Scripting, mshta, service execution, WMI, rundll32, and schedule tasks were used in the Execution step. In Persistence and Privilege escalation, no special technique was used. In the Defense evasion step, mshta, indicator removal on host technology was used. Credential dumping technology was used in the Credential access step, and file and directory discovery technology was used in the Discovery step. In the Lateral movement step, Petya used Windows admin shares and exploited remote services technology. In the Impact step, disk structure wipe and data encrypted for impact technologies were used.

**Table 10.** Scoring result of cyber-attacks using ATT&CK phases.

| No. | Cyber-Attack | Number of used techniques of MITRE ATT&CK | | | | | | | | | | | | Total |
|-----|--------------|---|---|---|---|---|---|---|---|---|---|---|---|-------|
| | | I | E | P | P | D | C | D | L | C | C | E | I | |
| 1 | Poweliks | 1 | 4 | 1 | 1 | 5 | - | - | - | 1 | 1 | 2 | - | 0.2581 |
| 2 | Rozena | 1 | 3 | - | 1 | 3 | - | - | - | - | 1 | - | - | 0.1452 |
| 3 | Duqu 2.0 | 1 | 2 | 1 | 2 | 1 | 1 | 4 | 2 | 1 | 1 | 1 | 1 | 0.2742 |
| 4 | Kovter | 1 | 4 | 1 | - | 5 | - | - | - | 1 | 1 | 1 | - | 0.2258 |
| 5 | Petya | 2 | 6 | - | - | 2 | 1 | 1 | 2 | - | - | - | 2 | 0.2581 |
| 6 | Sorebrect | 3 | 1 | - | 1 | 2 | 1 | 1 | - | - | 2 | - | 2 | 0.2097 |
| 7 | WannaCry | - | 1 | 2 | 1 | 2 | - | 2 | 2 | - | 2 | - | 3 | 0.2419 |
| 8 | Magniber | 1 | 4 | - | 1 | 1 | - | 1 | - | - | 1 | - | 2 | 0.1774 |
| 9 | Emotet | 1 | 4 | 1 | - | 2 | 3 | 1 | 1 | 2 | 5 | 1 | - | 0.3387 |
| 10 | GandCrab | 1 | 3 | - | 1 | 4 | - | 1 | - | - | 1 | 2 | 2 | 0.2419 |
| 11 | APT1 | - | 1 | - | - | 1 | 1 | 6 | 2 | 5 | - | - | - | 0.2581 |
| 12 | Emissary Panda | 1 | 4 | 3 | 5 | 6 | 3 | 6 | 2 | 9 | 2 | 1 | - | 0.6774 |
| 13 | APT29 | 2 | 6 | 2 | 3 | 3 | - | - | 1 | - | 3 | - | - | 0.3226 |
| 14 | Sector04 | 2 | 8 | - | - | 2 | 2 | - | - | - | 1 | - | 1 | 0.2581 |
| 15 | Lazarus Group | 2 | 4 | 3 | 3 | 11 | 2 | 8 | 2 | 6 | 7 | - | 7 | 0.8871 |
| 16 | APT38 | 1 | 1 | - | - | 4 | 1 | 2 | - | 2 | 1 | - | 7 | 0.3065 |
| 17 | Chafer | 2 | 4 | 3 | - | 1 | 2 | 3 | 2 | 1 | 1 | - | - | 0.3065 |
| 18 | MuddyWater | 2 | 8 | 1 | 1 | 4 | 6 | 6 | - | 2 | 6 | - | - | 0.5806 |

In contrast, Magniber, with a score of 0.1774, operates as follows. In the initial access step, drive-by compromise technology was used. In the Execution step, scripting, exploitation for client execution, scheduled task, and WMI were used. No technique seems to have been used in the Persistence step. Process injection technology was used in the Privilege escalation step, and obfuscated files or information technology was used in the Defense evasion phase. In the Discovery phase, files and directory discovery technology was used. No technique was used in the Lateral movement and Collection stages. In the Command and Control phase, remote file copy technology was used. In the Impact phase, data encrypted for impact and inhibit system recovery technologies were used.

The technique that was used for Emissary Panda (0.6774), which holds the highest score after Lazarus, is as follows. In the Initial access stage, the drive-by compromise technique was used. In the Execution phase, PowerShell, the windows command shell, Exploitation for client execution, and WMI were used. In the Persistence phase, registry run keys, create or modify system process, and web shell technologies were used. In the

Privilege escalation stage, bypass user access control, exploitation for privilege escalation, hijack execution flow, dll side-loading, and process hollowing techniques were used. In the Defense evasion phase, Windows event logging is disabled, file deletion, network share connection removal and obfuscated files or information were used. In the Credential access phase, OS credential dumping technique for LSA secrets, LSASS memory, and security account manager was used. In the Discovery phase, local account, network service scanning, and query registry methods were used. In the Lateral movement phase, the exploitation of remote services technique is used. In the Collection phase, automated collection is performed. In the Command and Control phase, web protocols, ingress tool transfer are used. In the Exfiltration step, archive via library is used.

Next, we analyzed APT38, which obtained a fairly low score; however, many techniques are used in the Impact step. APT38 used a drive-by compromise technique in the initial access step. In the Execution step, the Windows command shell was used. In the Defense evasion step, indicator removal on host, modify registry, and software-packing techniques were used. In the Credential access step, input capture technique was used. In the Collection step, clipboard data were executed. In the Command and Control step, web protocols and ingress tool transfer were used. In particular, in the Impact step, many techniques were used: data destruction, data encrypted for impact, data manipulation, disk structure wipe, and system shutdown techniques.

### 4.4.3. Scoring Result Summary

We derived the score for the final cyber-attacks by combining the Cyber Kill Chain score including cybersecurity offensive elements and the score based on MITRE ATT&CK as shown in Figure 7. Each fileless cyber-attack score shows powerliks (0.6295), Rozena (0.4881), Duqu 2.0 (0.6171), Kovter (0.5687), Petya (0.6867), Sorebrect (0.4954), WannaCry (0.6133), Magniber (0.4631), Emotet (0.6816), Gandcrab (0.5848). We can show that Petya has the highest score of the fileless cyber-attacks. Each APT cyber-attack score shows APT1 (0.6581), Emissary Panda (1.1631), APT29 (0.7512), Sectorj04 (0.7152), Lazarus Group (1.4014), APT38 (0.7351), Chafer (0.7065), Muddywater (1.0092). The Lazarus group APT shows the highest score of the cyber-attacks. Overall, APT cyber-attacks usually score higher than fileless cyber-attacks, because he APT use more ATT&CK techniques. The limitation of our approach is that real malware cannot be analyzed. However, we believe that measuring the score of cyber-attacks is meaningful as an initial research step.

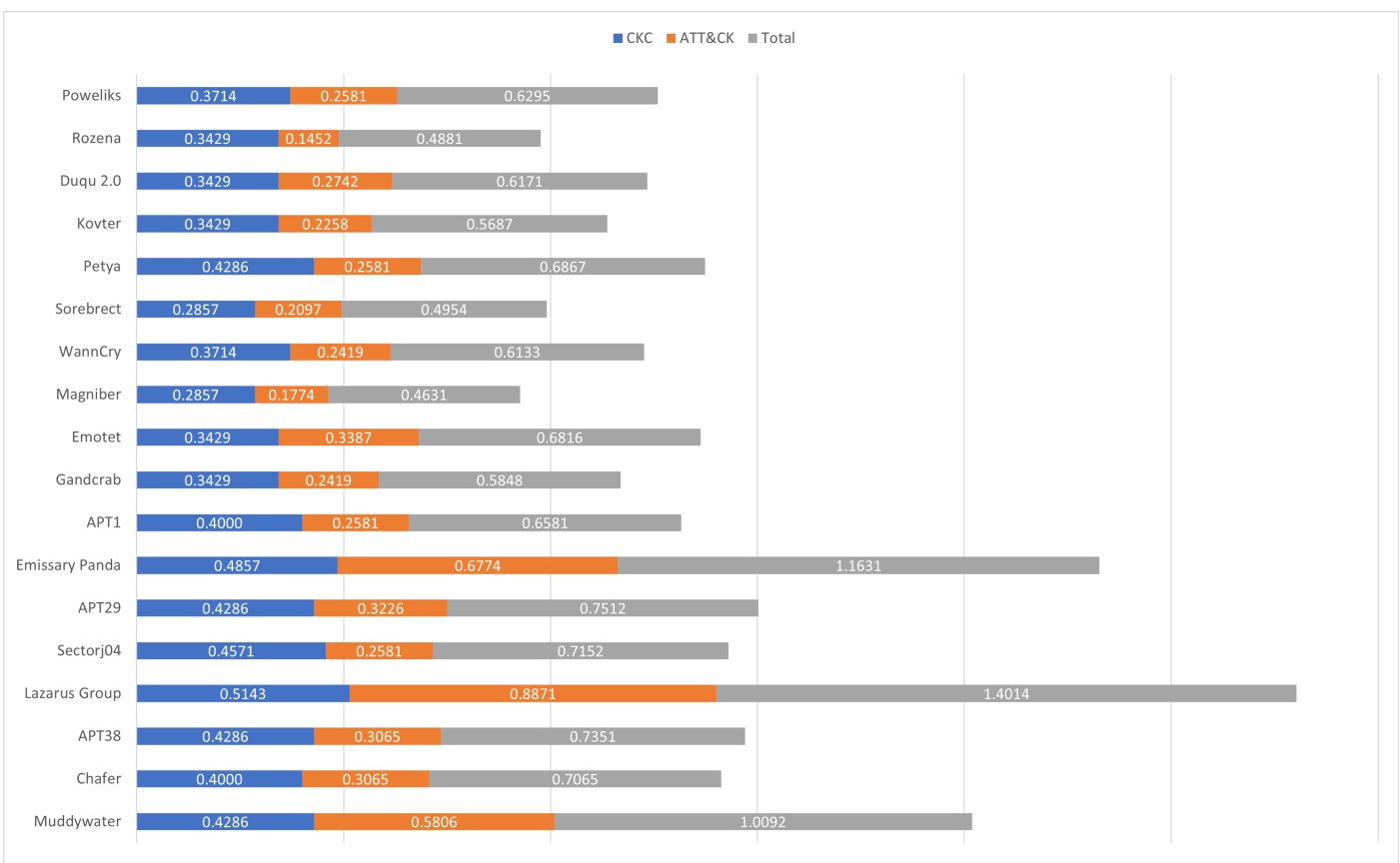

**Figure 7.** Scoring total result for cyber-attacks.

## 5. Conclusions

Cyber-attacks are constantly evolving. Starting with simple attacks, such as the defacement of a homepage or the acquisition of personal information, cyber-attacks have been changing to become more complex, such as the theft of national secrets and attacks on national infrastructure. Cyber-attacks include various attack elements. Much security research has addressed the sophistication of cyber-attacks; however, research on scoring attack complexity has been lacking. We conducted research on the complexity of attacks and proposed a model for offensive cybersecurity. In addition, important elements of the offensive cybersecurity model were identified, and detailed descriptions of each element were investigated and described.

Based on this study, we derived scores for fileless and APT group cyber-attacks. The results can be quantified for each of the various elements and techniques of each cyber-attack. The method we investigated was scored on the basis of public cyber-attack reports. This study is the first to be conducted to quantify and score cyber-attacks. We found that APT cyber-attacks have higher scores than fileless cyber-attacks, due to the APT using various ATT&CK techniques. In future research, we will expand to adopt automatic report analysis and gather input from more expert focus groups. In the future, we believe that many researchers are expected to be able to contribute to safeguarding cyberspace from cyber-attacks by researching and developing measurable scoring models for cyber-attacks through our initial research.

**Author Contributions:** Conceptualization, K.K.; methodology, K.K.; software, K.K.; validation, K.K., F.A.A. and H.K.; formal analysis, K.K.; investigation, K.K. and F.A.A.; resources, K.K., F.A.A. and H.K.; data curation, K.K.; writing—original draft preparation, K.K.; writing—review and editing, F.A.A. and H.K.; visualization, K.K.; supervision, H.K.; project administration, K.K.; funding acquisition, H.K. All authors have read and agreed to the published version of the manuscript.

**Funding:** For Kim, K, and Alfouzan, F.A, this research received no external funding. For Kim H.K, this research was funded by the Korea government (MSIT).

**Acknowledgments:** This research was conducted during the work as an Assistant Professor at Naif Arab University for Security Sciences (NAUSS), for Kyounggon Kim and Faisal Abdulaziz Alfouzan. Furthermore, We would like to express our sincere gratitude to Naif Arab University for Security Sciences (NAUSS) and the president of the university for his consistent support and encouragement. For y Huy Kang Kim, this work was supported by Institute of Information & communications Technology Planning & Evaluation (IITP) grant funded by the Korea government (MSIT) (No. 2021-0-00624, Development of Intelligence Cyber Attack and Defense Analysis Framework for Increasing Security Level of C-ITS).

**Conflicts of Interest:** The authors declare no conflict of interest.

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
