# Peer review of "Cyber-Attack Scoring Model Based on the Offensive Cybersecurity Framework"

_applsci, doi:10.3390/app11167738_

Round 1
Reviewer 1 Report
This paper identified each element of offensive cyber-security used in cyber-attacks and investigated extent to which the detailed techniques identified in the offensive cyber-security framework were used by analyzing cyber-attacks.
Good point:
+ Some results have shown.
Major issues:
- The contributions of the paper are not clear. The authors needs to clarify what the main methods used in your framework.
- It should add some detail description for each figure and tables such as Figure 1 and 2.
- The author should compare the proposed approach with other similar works or provide a discussion. Otherwise, it's hard for reader to identify the novelty and contribution of this work.
- Adjust the format of the figures such as Figure 1 and 2. In addition, check the number of the sections.
- Language: the English needs to improved, as there are many typos and unclear sentences, which is not easy for readers to understand.
For example, “It also that the nature of cyber-attacks is gradually changing.”
“Many researches regarding cyber-attacks has been conducted, however there has been lack of research related to measure for cyber-attacks perspective using offensive cybersecurity.” has-->have
“This motivated us to propose a methodology for quantifying cyber-attacks such that they are measurable rather than abstract.” such that --> so that
Please check the whole paper.
Author Response
Dear respected reviewers,
Please find the attached file which includes our response to your comments.
Thanks

Reviewer 2 Report
The authors propose a methodology for quantifying cyber attacks. The idea is acceptable; but the effectiveness and the application of such a score is not convincing from the article. The quality of writing could have been better; there are too many things in the article and not so easily comprehendible. There is a kind of survey of attacks being done in the first part of the paper. Some of the concepts/techniques seems vague.
- What is the purpose of proposing such a score?
- Measuring cyber-attack - with this score what conclusions can be driven about an attack?
- In what stage does this score be used? Will this be useful for organizations?
- Is this a once-time process? If a cyber attack happens anywhere, in what way will this score help?
Author Response
Dear respected reviewer,
Please find the attached file which includes our response to your comments.
Thanks

Round 2
Reviewer 1 Report
No more comments
Author Response
Kindly find the attached file.

Reviewer 2 Report
Still have the same queries that are already raised. The authors could have highlighted the changes and address the comments so that its easily understandable. From comparing the files, there are some edits done.
The practical usage of such a score is not so impressively presented. This can be considered as a survey paper. But the second part demands a bit more explanation with convincing results
Author Response
Kindly find the attached file.
